# Study of the Effect of PGDA Solvent on Film Formation and Curing Process of Two-Component Waterborne Polyurethane Coatings by FTIR Tracking

**Ruitao Wang [1], Chunxiang Li [1], Zhigang Liu [1], Zhongping Yao [1], Zhijiang Wang [1], Zhaohua Jiang [1],* and Min Zhao [2]**

[1] School of Chemistry and Chemical Engineering, Harbin Institute of Technology, Harbin 150001, China; wruitao@126.com (R.W.); lichx@hit.edu.cn (C.L.); lzgbs@hit.edu.cn (Z.L.); yaozhongping@hit.edu.cn (Z.Y.); wangzhijiang@hit.edu.cn (Z.W.)

[2] CRRC Qingdao Sifang Co. LTD., Qingdao 266111, China; zhaomin@cqsf.com

\* Correspondence: jiangzhaohua@hit.edu.cn

**Abstract:** Waterborne polyurethane coatings were prepared using polyhydroxyacrylate dispersion, polyisocyanate, and propylene glycol diacetate (PGDA). The rate of reaction between hydroxyl and NCO groups in film formation and curing processes was studied by FTIR. The influence of PGDA amount on film formation and curing was also studied. Results showed that PGDA content had a significant effect on the curing process. With less than 10% PGDA, the role of PGDA was more to assist diffusion of polyhydroxyacrylate and polyisocyanate resin droplets. This promoted the reaction between hydroxyl groups and NCO. With more than 10% PGDA, its effect as a solvent was more and it inhibited the reaction between hydroxyl and NCO groups. When the amount of PGDA was about 10%, the synergy between both the roles promoted the crosslinking and curing reactions. The extent of the curing reaction of NCO was more than 70% in 4 h, which was significantly higher, compared with that of about 30% without PGDA. The good applicability and appearance of the waterborne polyurethane coating prepared in this study were verified for the application to carbon fiber metro vehicles.

**Keywords:** waterborne polyurethane coatings; film formation; FTIR tracking

---

## 1. Introduction

Polyurethane coating, owing to its highly decorative and weather-resistant features, is an important asset to the coating industry and is widely used for coating wood, automobiles, and rail transit vehicles [1,2]. Waterborne polyurethane coating is environmentally friendly and is made up of two components: base and hardener. The main components of the base are OH functional acrylate dispersion, pigment, additives, and water. The hardener is mainly composed of polyisocyanate resin and alcohol ether solvent. During application, the base and hardener are mixed and stirred mechanically to disperse the polyisocyanate in water containing the base to obtain an emulsion. After evaporation of water, the polyisocyanate and hydroxy resins fuse and crosslink, to form a polymer and are then cured to form a film. The basic reaction is as follows [3]:

$$R-NCO + R'OH \longrightarrow R'O-\overset{\overset{\textstyle O}{\|}}{C}-NH-R \tag{1}$$

However, due to the high reactivity of the NCO groups in polyisocyanate, side reactions occur during the mixing of the base and hardener, which are as follows:

$$R-NCO + H_2O \longrightarrow R'NH-\overset{\overset{\textstyle O}{\|}}{C}-OH \longrightarrow R-NH_2 + CO_2 \tag{2}$$

$$R-NCO + R'-NH_2 \longrightarrow R'NH-\overset{\overset{\textstyle O}{\|}}{C}-NH-R \tag{3}$$

In Reaction (2), the NCO group reacts with water producing $CO_2$ gas. During the curing of the coating, the initial reaction between NCO groups and water molecules is carried out in a large amount of water. At this time, since the coating does not form a film and the viscosity is low, the $CO_2$ gas formed can escape quickly. When most of the water volatilizes, the viscosity of the coating increases rapidly. The $CO_2$ gas produced in the reaction between traces of water in the coating and NCO groups escapes with difficulty, resulting in the formation of microbubbles in the coating. The microbubbles have a negative impact on the appearance and protective performance of the film [4]. How to solve the problem of $CO_2$ emission in the chemical reaction Equation (2) is the key to inhibit the formation of microbubbles in the coating.

Water in the paint evaporates after the application of the coating. Thereafter, the coating undergoes change from the emulsion phase to the solution state. This solution system is based on alcohol ether solvent as medium and OH functional acrylate resin and polyisocyanate resin as the solutes. At this time, the coating should have good fluidity to allow the easy escape of $CO_2$ gas. This prevents the gas from remaining inside the coating or forming microbubbles on the surface. Generally, a certain amount of alcohol ether solvent is added to the mixture, to reduce the viscosity of the resin, and help $CO_2$ to escape. This is termed "opening time" in the coating [5].

Researchers in the coating industry have conducted some practical research on the types and amounts of these alcohol ether solvents used in waterborne polyurethane coatings. According to Wicks, the polyisocyanate component in waterborne polyurethane coatings is sometimes diluted with a solvent to reduce its viscosity [6]. The waterborne polyurethane coating prepared by Cakic showed satisfactory pot-life and hardness on the addition of about 10% solvent [7]. Yin et al. synthesized a new kind of green and functional two-component flame-retardant waterborne polyurethane and its coating. It was prepared by adding polyurethane polyol dispersions, hydrophilic nano $TiO_2$, hydrophilic curing agent, and alcohol ether solvents. These green and functional two-component waterborne polyurethane coatings had an excellent appearance and tensile properties [8]. Deyong studied the effect of propylene glycol diacetate (PGDA) solvent on film formation and found that higher boiling point of alcohol ether solvent and longer opening time of the paint film contributed to the timely discharge of bubbles, caused by physical agitation and chemical reactions [9].

All these studies have shown that a larger amount of solvent, such as PGDA, favored the emission of $CO_2$. The reason as to why PGDA can be used as a solvent in the two-component polyurethane coating is that its structure does not contain hydroxyl groups. The solvent containing hydroxyl functional groups affects the network formation of the polymer, causing drastic reduction of the mechanical stability (hardness, tensile strength, etc.) of the coating. Although PGDA does not contain OH groups in its structure that can react with NCO group, it can volatilize completely during the film-forming process and does not affect the final performance of the film, but since the boiling point of PGDA touches 161 °C, the volatilization rate is far less than that of water. This increases the drying time of the film and consequently reduces its applicability. Moreover, PGDA belongs to the class of volatile organic compounds (VOCs) [10]. The excessive addition of PGDA violates the norms of environmental friendliness of waterborne coatings, and hence it is necessary to determine the appropriate amount of PGDA required.

According to Equation (2), $CO_2$ is the product of the reaction of isocyanate with water. To obtain a film without microbubbles, the opening time for the escape of $CO_2$ gas during the film-forming

process should match with the time of reaction that produces $CO_2$ gas. Shorter the chemical reaction time, the shorter is the film opening time required, the shorter is the film drying time, and the higher is the application efficiency of the coating, which is desirable. Although PGDA does not participate in the chemical reaction, in a multi-phase system, its effect on the chemical reaction is quite complex. It promotes the fusion of reactant resin particles and also has a solvent effect on the reaction.

The effect of a catalyst on the reaction of polyurethane and the role of a solvent on the reaction of epoxy polymer were studied [11,12]. However, only a few pieces of theoretical research have focused on the role of PGDA in the reactions of polyurethane coatings. Nevertheless, it is significant as a universal guide for the formulation and design of two-component waterborne coatings.

Ensuring that the film layer is free of microbubbles is of great significance for improving the application efficiency of the coating. Clarity on the influence of alcohol ether solvent on the chemical reaction during the film formation and curing process of two-component waterborne polyurethane coatings helps to control the quality of the coating. It also helps to seek a relationship between the reactivity of NCO groups and the opening time of the film.

FTIR is a powerful tool to study reaction rate and kinetics. In the study of Fei et al., FTIR is used to monitor the reaction between phenol and tolylene-2,4-diisocyanate in different polar solvents by the intensity of NCO absorbance. They carefully examined the relationship between absorbance (2273 $cm^{-1}$) and concentration of NCO in solvents [13].

In this paper, two-component waterborne polyurethane coatings were prepared using polyhydroxyacrylate dispersion, polyisocyanate resin, and PGDA, which are commonly used in the coating industry. The influence of PGDA content on the chemical reaction was studied by Fourier transform infrared spectroscopy (FTIR). The effect of PGDA content on the progress of the NCO reaction was discussed.

## 2. Materials and Methods

### 2.1. Materials

The polyhydroxyacrylate dispersion (Bayhydrol A 2470) was purchased from Covestro Co. (Barcelona, Spain), and its physical and chemical properties are listed in Table 1.

**Table 1.** Physical and chemical properties of polyhydroxyacrylate dispersion Bayhydrol A 2470.

| Property | Value | Unit of Measurement | Method |
|---|---|---|---|
| Viscosity at 23 °C, $D$ = approx. 40 $s^{-1}$ | 1500–3000 | mPa·s | DIN EN ISO 3219/A.3 [14] |
| Non-volatile content (1 g/1 h/125 °C/convection oven) | 44–47 | % | DIN EN ISO 3251 [15] |
| Hydroxyl content (solvent-free, calculated) | Approx. 3.9 | % | - |
| Hydroxyl equivalent | 979.2 | - | - |

Bayhydur XP 2655 was the polyisocyanate reagent chosen as the curing agent and was procured from Covestro Co. (Domagen, Germany). It is a hydrophilic sulfonate modified hexamethylene diisocyanate (HDI) trimer. The physical and chemical properties are presented in Table 2.

**Table 2.** Physical and chemical properties of polyisocyanate Bayhydur XP 2655.

| Property | Value | Unit of Measurement | Method |
|---|---|---|---|
| Viscosity at 23 °C | 3500 ± 1000 | mPa·s | DIN EN ISO 3219/A.3 [14] |
| NCO content | 20.3–21.3 | % | DIN EN ISO 11909 [16] |
| NCO equivalent | - | 201.9 | - |

Dawanol PGDA, obtained from Dow Co. (Midland, MI, USA), was chosen as the solvent and its physical and chemical properties are shown in Table 3.

**Table 3.** Physical and chemical properties of Dawanol propylene glycol diacetate (PGDA).

| Property | Value | Unit of Measurement | Note |
|---|---|---|---|
| Molecular weight | 160 | g/mol | - |
| Boiling point | 161 | °C | @760 mm Hg |
| Flash point | 86 | °C | - |
| Evaporation rate | 3.6 | - | Butyl acetate = 100 |
| Solubility | 7.4 | - | 25 °C in water |
| Solubility | 4.1 | - | 25 °C water-soluble in |

The structural formula of PGDA is shown in Figure 1.

**Figure 1.** Structural formula of PGDA

### 2.2. Pretreatment of the Resin Coating

In the formulation design of waterborne polyurethane coatings, NCO/OH molar ratio is often higher than 1, since side reactions consume some portion of NCO groups [17]. However, in this study, the reaction rate was monitored by the reduction in the number of NCO groups. Hence, in order to facilitate theoretical analysis, the equivalent ratio of NCO/OH in this study was chosen as 1/1 (weight ratio of Bayhydrol A 2470/Bayhydur XP 2655 taken was 4.85/1). According to this ratio, polyhydroxyacrylate dispersion and polyisocyanate curing agent were mixed, a certain amount of PGDA solvent was then added and the entire mixture was mechanically stirred for 3 min to mix evenly.

In order to make the infrared tracking spectrograms more consistent at different periods and eliminate the errors in the results caused due to different test positions, each test sample was placed in the infrared chromatograph for the entire tracking time till the end of the test.

A 100 μm wire bar was used for preparing the film on a glass plate. The film was placed under the standard conditions of 25 °C and 50 RH for 15 min. After most of the water evaporated, the film changed from water phase to oil phase and was subsequently tested.

### 2.3. Characterization of Typical Functional Groups and Quantitative Analysis of Curing Reaction

The reaction samples prepared on the glass plate were put into the infrared spectrum tester (Thermo Fisher Scientific, Shanghai, China). The FTIR analysis was conducted in the ATR mode on a Thermo Fisher Nicolet iS10 instrument. The spectrum analysis was carried out every 1 h, the infrared spectrum was recorded. The reaction progress was monitored by analyzing the changes in functional groups. The reaction degree curve was drawn and the effect of varying PGDA content on the rate of the chemical reaction during curing was analyzed.

In order to normalize the data, the absorption peak area of NCO was compared with the peak at 2950–2850 cm$^{-1}$ (C–H alcane stretching) which was an inert group in the reaction, and the change of the ratio was used to characterize the reaction degree of the NCO group. In theory, the peak of 2950–2850 cm$^{-1}$ will also be affected by the volatilization of PGDA due to its C–H bond. However, considering that the boiling point of PGDA is 161 °C, the test temperature is 25 °C, and the coating has been in the closed environment of the infrared spectrum tester within 4 h of the test process, the evaporation is very limited, so this factor is ignored in this study.

According to Yang's research results, there are many factors influencing the establishment of the linear relationship between the absorption peak area and NCO concentration. They found that when the concentration of NCO in four different solvents was 0–0.4 mol L$^{-1}$, the linear relationship

between NCO concentration and its absorbance was correct. When the concentration exceeded, their relationship was no longer a strict linear relationship [18]. However, one thing is very certain: NCO absorption will increase with the increase of NCO concentration. Although this relationship is not enough to study the reaction kinetics accurately and quantitatively, it is feasible to study the effect of solvent on the reaction process to promote or inhibit.

## 3. Results and Discussion

Physical and chemical changes occur during the curing process and film formation, involving a two-component waterborne polyurethane coating. Moreover, the physical factors also have an effect on the chemical changes of NCO and OH groups. Therefore, on the premise that no microbubbles are formed in the film, the NCO conversion reaction rate for different amounts of PGDA was measured.

### 3.1. FTIR Analysis of Curing Process

PGDA solvent was added in proportions of 0%, 5%, 10%, 15%, and 20% of the total solid weight of polyhydroxyacrylate dispersion and polyisocyanate resin. The ratios of test samples are presented in Table 4. The chemical reaction between the hydroxyl and NCO groups of five test samples from 15 min to 4 h was tracked by infrared spectroscopy.

**Table 4.** Ratio of reactants for the curing process.

| Sr. No. | Bayhydrol A 2470 | Bayhydur XP 2655 | PGDA | PGDA/Resin Solid |
|---------|------------------|------------------|------|------------------|
| 1 | 48.50 | 10.00 | 0.00 | 0/100 |
| 2 | 48.50 | 10.00 | 1.56 | 5/100 |
| 3 | 48.50 | 10.00 | 3.12 | 10/100 |
| 4 | 48.50 | 10.00 | 4.68 | 15/100 |
| 5 | 48.50 | 10.00 | 6.24 | 20/100 |

Films of the five samples with different amounts of PGDA were visually observed. The surfaces of the films with 0% and 5% PGDA showed microbubbles, whereas the films with 10%, 15%, and 20% PGDA addition were smooth and dense, without microbubbles or other defects.

Figure 2 is the spectra showing reaction progress for different contents of PGDA.

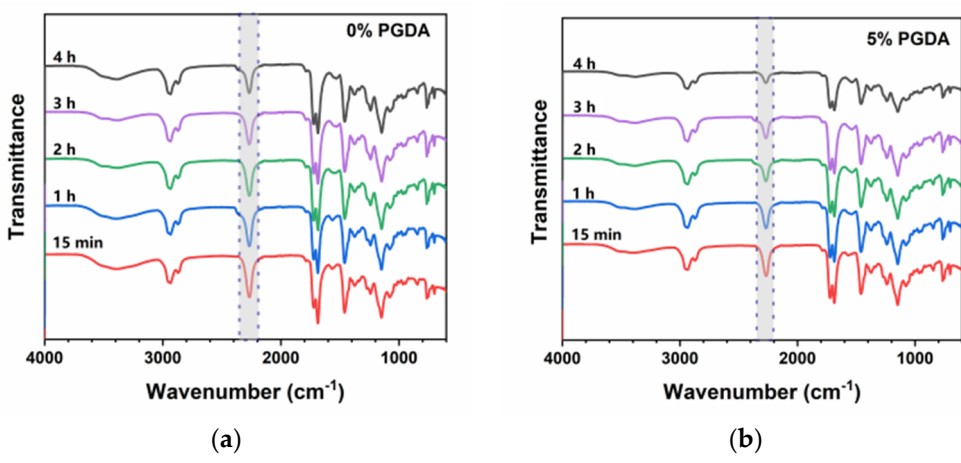

(a)    (b)

**Figure 2.** *Cont.*

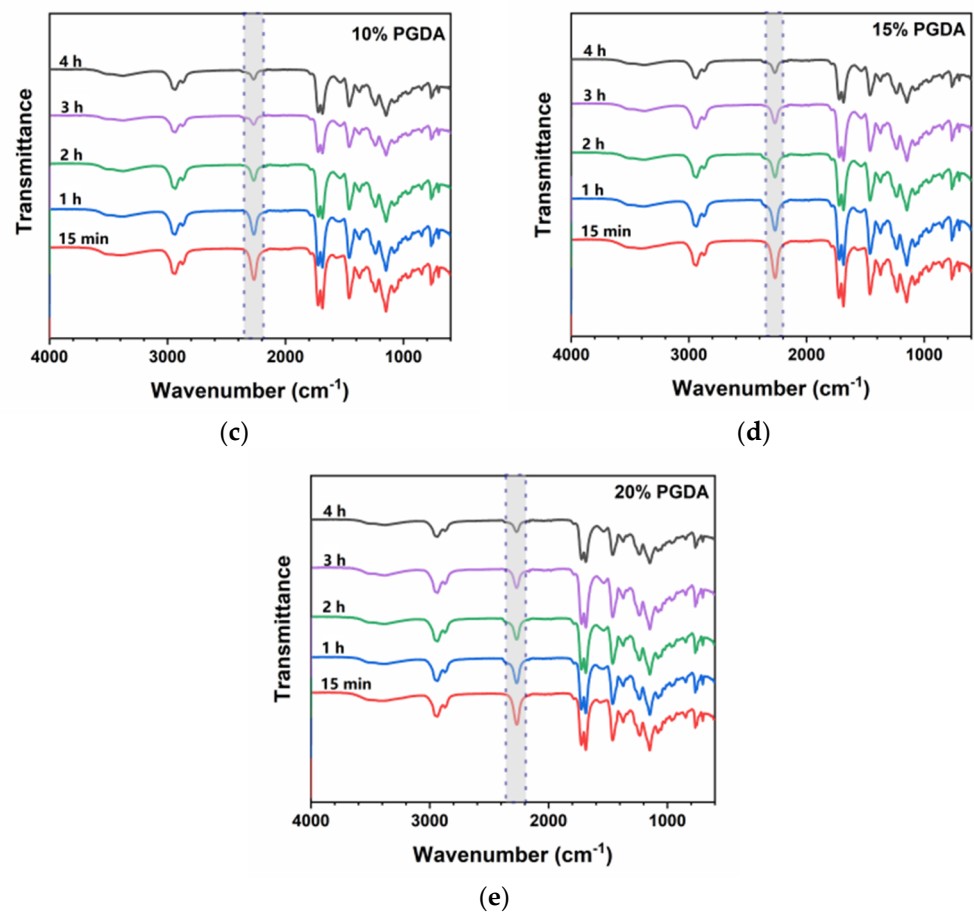

**Figure 2.** Progress of the curing reaction by FTIR spectroscopy (**a**) 0% PGDA; (**b**) 5% PGDA; (**c**) 10% PGDA; (**d**) 15% PGDA; (**e**) 20% PGDA.

The progress of the reaction was followed by changes in the intensities of the peaks for NCO, hydroxyl, and urethane functional groups. According to Reactions (1)–(3), the gradual decrease in the absorption peak of the NCO group at 2270 cm$^{-1}$ in the infrared spectrum is attributed to its reaction with hydroxyl groups and water resulting in the formation of carbamate and urea functional groups.

The peaks for carbonyl carbamate formed in Reaction (1) appeared at 1702–1740 cm$^{-1}$, and the absorption peak of urea carbonyl generated by Reactions (2) and (3) appeared at 1630–1689 cm$^{-1}$ [19].

From the above figures, it was evident that the intensity of the urea carbonyl peak showed no obvious changes for different concentrations of PGDA. This showed that PGDA did not selectively inhibit or promote the side reactions shown in reaction Equations (2) and (3) alone. The absorption peak at 2270 cm$^{-1}$ is characteristic of the NCO group. As the reaction progressed, the peak intensity obviously decreased. Moreover, the rate of change of NCO peak was not proportional to the amount of PGDA.

According to the Beer–Lambert law [20],

$$A = \log(1/T) = Kbc \qquad (4)$$

*A* is the absorbance; *T* is the transmittance, which is the ratio of the outgoing light intensity (*I*) to the incident light intensity (*I$_0$*); *K* is the molar absorption coefficient, which is dependent on the nature of the absorbing material and the wavelength of the incident light $\lambda$; *C* is the concentration of the absorption material, in mol/L; *B* is the thickness of the absorption layer, in cm.

Since absorbance A is directly proportional to peak area $S$, the degree of curing reaction of NCO ($P_{NCO}$) can be calculated as follows:

$$P_{NCO} = 1 - \frac{A'_{NCO}/A'_{CH}}{A^0{}_{NCO}/A^0{}_{CH}} = 1 - \frac{S'_{NCO}/S'_{CH}}{S^0{}_{NCO}/S^0{}_{CH}} \tag{5}$$

$A^0{}_{NCO}$ and $A'_{NCO}$ are the absorbances of NCO peak (2270 cm$^{-1}$) at the initial time and time $t$, respectively; $A^0{}_{CH}$ and $A'_{CH}$ are the absorbances of CH peak (2950 cm$^{-1}$); at the initial time and time t, respectively; $S^0{}_{NCO}$ and $S'_{NCO}$ are the absorption peak areas of NCO at the initial time and time t, respectively; $S^0{}_{CH}$ and $S'_{CH}$ are the absorption peak areas of CH at the initial time and time t, respectively.

From Figure 2 and Equation (5), the extent of the curing reaction of NCO ($P_{NCO}$) with different contents of PGDA could be calculated at different reaction times, and the values are presented in Table 5.

**Table 5.** Characteristic peak areas and reaction degrees of NCO.

| PGDA Content | 0% | | | 5% | | | 10% | | | 15% | | | 20% | | |
| Reaction Time | $S_{NCO}$ | $S_{CH}$ | $P_{NCO}$ | $S_{NCO}$ | $S_{CH}$ | $P_{NCO}$ | $S_{NCO}$ | $S_{CH}$ | $P_{NCO}$ | $S_{NCO}$ | $S_{CH}$ | $P_{NCO}$ | $S_{NCO}$ | $S_{CH}$ | $P_{NCO}$ |
|---|---|---|---|---|---|---|---|---|---|---|---|---|---|---|---|
| 15 min | 2342 | 1739 | 0.0% | 2177 | 1642 | 0.0% | 1847 | 1409 | 0.0% | 2190 | 1721 | 0.0% | 2028 | 1641 | 0.0% |
| 1 h | 2327 | 1756 | 1.6% | 1842 | 1656 | 16.1% | 1340 | 1413 | 27.7% | 1946 | 1732 | 11.7% | 1810 | 1635 | 10.4% |
| 2 h | 1921 | 1738 | 17.9% | 1746 | 1653 | 20.3% | 929 | 1417 | 50.0% | 1484 | 1722 | 32.3% | 1515 | 1632 | 24.9% |
| 3 h | 1698 | 1753 | 28.1% | 1560 | 1658 | 29.1% | 544 | 1276 | 67.5% | 1016 | 1729 | 53.8% | 1188 | 1636 | 41.2% |
| 4 h | 1608 | 1742 | 31.5% | 980 | 1434 | 48.5% | 544 | 1389 | 70.1% | 797 | 1668 | 62.4% | 870 | 1453 | 51.5% |

Using the data in Table 5, the extent of reaction of the NCO group $P_{NCO}$ with 0%, 5%, and 10% PGDA can be depicted, as shown in Figure 3.

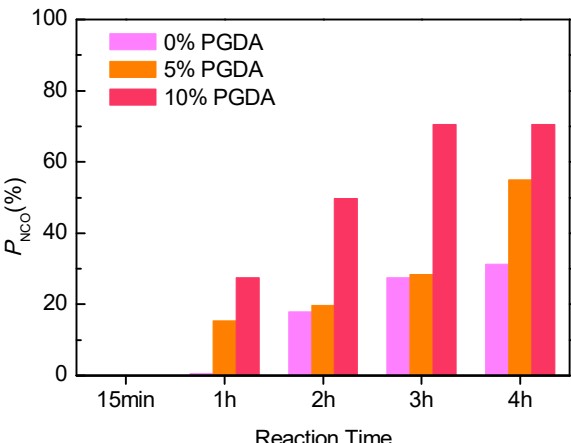

**Figure 3.** Extent of NCO reaction during the curing process of waterborne polyurethane coating (0%, 5%, 10% PGDA).

In addition, using the data in Table 5, the extent of reaction of the NCO group when the PGDA content is 10%, 15%, and 20%, is shown in Figure 4.

The effect of different PGDA contents on the chemical reaction can be studied from the figures. In Figure 3, as the amount of PGDA increased from 0% to 10%, the rate of NCO consumption increased gradually and thus the rate of chemical reaction rate also increased. In Figure 4, when the amount of PGDA was increased from 10% to 20%, the rate of NCO consumption decreased gradually with an increase in PGDA content and the chemical reaction rate decreased gradually. Hence, 10% of PGDA was found to be the best for promoting crosslinking and curing reactions.

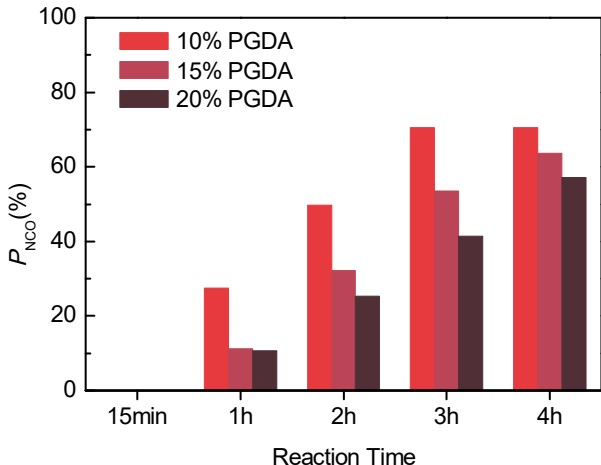

**Figure 4.** Extent of NCO reaction during the curing process of waterborne polyurethane coating (10%, 15%, 20% PGDA).

For seven days' drying, the morphology of each test coating was observed under an optical microscope, and the results are shown in Figure 5. The surface of Sr. No. 1 sample in Table 4 (0% PGDA) showed obvious microbubbles. Few microbubbles were seen in the sample of Sr. No. 2 (5% PGDA). There were no microbubbles on the surfaces of Sr. Nos. 3–5 (10–20% PGDA). This shows that a concentration of more than 10% PGDA was effective in eliminating the microbubbles.

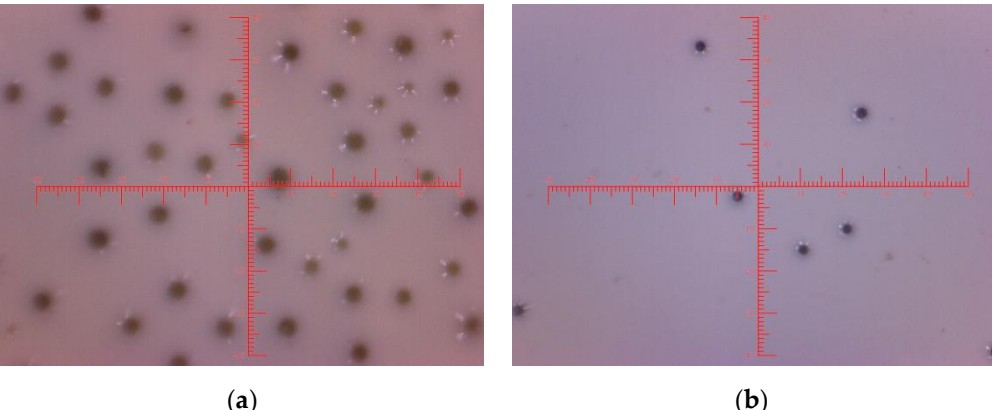

|     (a)     |     (b)     |

**Figure 5.** Appearance of coating surfaces under an optical microscope at 200× magnification (**a**) Sr. No. 1; (**b**) Sr. No. 2.

### 3.2. Role of PGDA in Assisting the Diffusion between polyhydroxyacrylate dispersion and polyisocyanate

In the process of film formation and curing of a two-component waterborne polyurethane coating, a mutual diffusion occurs between polyhydroxyacrylate resin particles and polyisocyanate resin droplets. The presence of a co-solvent can reduce the $T_g$ of the resin and help the fusion process. Complete fusion of polyhydroxy acrylate resin and polyisocyanate can improve the contact between hydroxyl groups and NCO groups and thereby increase the reaction rate. The diffusion and fusion processes are depicted in Figure 6, respectively.

In absence of co-solvent or very low amount of co-solvent, as in the case of Sr. No. 1 in Table 4 (0% PGDA), the fusion of the polyhydroxyacrylate resin particles and polyisocyanate resin droplet is expected to be poor. This reduces the reaction rate and results in an inadequate reaction, as shown in Figure 7.

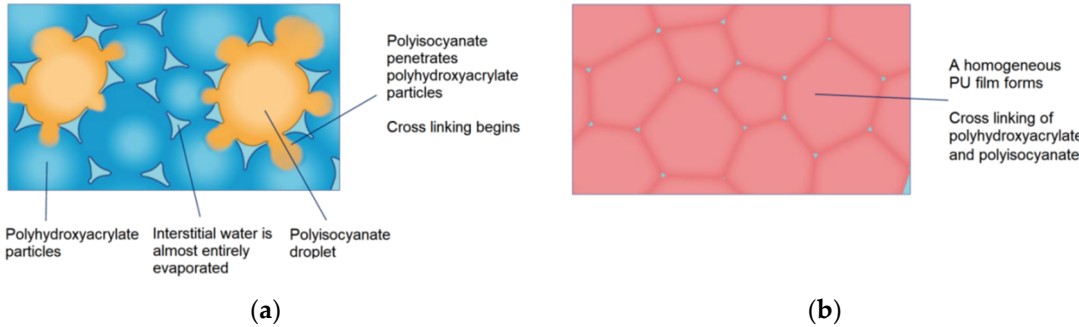

**(a)**　　　　　　　　　　　　　　　　　　　　**(b)**

**Figure 6.** The process of film formation and curing of a two-component waterborne polyurethane coating. (**a**) Process of diffusion between polyhydroxyacrylate particles and polyisocyanate droplets; (**b**) Chemical crosslinking in fully diffused and fused resin.

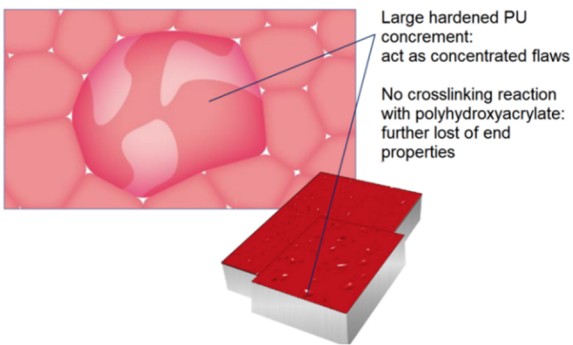

**Figure 7.** Chemical crosslinking reaction in resin with poor diffusion and fusion.

From the data of Figure 3 for the reaction when the PGDA content was less than 10%, the reaction rate increased with an increase in the PGDA amount. This showed that the addition of PGDA could promote the diffusion of resin particles participating in the chemical reaction, increase the contact area of reaction groups, and thus promote the reaction.

This can also be explained by the collision theory of Trautz and Lewis [21]. The reaction between hydroxyl and isocyanate groups requires the reaction particles to get closer to each other and the reaction rate is proportional to the number of collisions. The addition of an appropriate amount of PGDA can facilitate the fusion of two resins and consequently increase the chances of mutual collisions between the two functional groups, thereby increasing the reaction rate.

### 3.3. Role of PGDA as Solvent in the Reaction between Hydroxyl and NCO Groups

When the amount of PGDA was more than 10%, the reaction rate began to decrease with the increase of PGDA. PGDA is a polar solvent. The effect of solvent polarity on reaction rate is explained by the Houghes–Ingold rule [22].

Houghes and Ingold used transition state theory to determine the effect of solvent on the reaction rate. Most of the transition states generated by the interactions between reactants are dipolar activated complexes, which are often significantly different from the corresponding starting reactants in charge distribution. Based on this, the following two rules are summarized.

For the reaction with an increase in charge density from the initial reactant to activated complex, the polarity of solvent increases, which is unfavorable for the complex formation and the reaction rate, is slowed down.

For a reaction with little change of charge density from the initial reactant to the activated complex, change in solvent polarity has little effect on the reaction rate.

Although the above rules have some limitations, they can be used to predict the solvent effects in transition states of many dipole reactions, such as electrophilic and nucleophilic substitution reactions,

P elimination reactions, and electrophilic addition reactions of unsaturated systems, and many experimental data were obtained [23].

The transition state and final product of the chemical reaction between the hydroxyl group and isocyanate group can be shown by Reaction (6):

$$R-N=C=O + R'OH \longrightarrow R-\bar{N}=\overset{+}{C}=\bar{O} \longrightarrow \left[ R-N=\underset{OR'}{\overset{|}{C}}-OH \right] \longrightarrow R-\underset{OR'}{\overset{H}{\underset{|}{N}}}-\overset{|}{C}=O$$

(6)

Prior to the reaction, oxygen and nitrogen atoms on the NCO group are electronegative, while the electron density at the carbon atom is low and positive. Since it is electrophilic, it can be readily attacked by nucleophiles and the chemical reaction occurs. However, the oxygen atom in the hydroxyl group with high electron density first reacts with the electropositive carbon atom to form a transition state. At this time, the charge density of the activated complex increases, which is consistent with the first case of the Houges–Ingold rule. When the amount of PGDA is more than 10%, its role as a solvent begins to appear. With an increase in the PGDA amount, the reaction rate decreases. Meanwhile, the effect of solvent polarity is not obvious when the amount of PGDA is less. However, it can facilitate the fusion of two resin particles with each other, increase the chances of collisions between two reactant groups, and increase the reaction rate.

*3.4. Industrial Application*

This study showed that when the amount of PGDA was about 10% of resin solid, it could promote the chemical crosslinking reaction most obviously, and was also more favorable for drying of the film. This ratio served as a reference for the formulation of a two-component waterborne polyurethane coating. Using the reactant ratios mentioned in Sr. No. 3 in Table 4, we prepared a two-component waterborne polyurethane coating, which met with the performance requirements of CRRC for polyurethane coatings and was applied on to a new generation of the metro vehicle with carbon fiber composite of CRRC. After application, the drying speed of the coating was fast, wherein the surface dried at room temperature within 30 min. Moreover, the surface of the coating was even, the appearance was excellent, and there were no microbubbles or any other defects.

## 4. Conclusions

When the amount of PGDA was less than 10%, its role in assisting diffusion was greater than that as a solvent. The test results showed that PGDA played a significant role in promoting the reaction. When the dosage was more than 10%, its role as a solvent was greater than that in assisting diffusion and so inhibited the reaction. Hence, 10% of PGDA of the solid resin was best for promoting the crosslinking and curing reactions. When the reaction time was 4 h, the extent of the curing reaction of NCO reached more than 70%.

Since the diffusion of resin particles and solvation occurs at the same time, the effect of PGDA on the reaction rate of hydroxyl and isocyanate groups in the curing process of a two-component waterborne polyurethane coating is complex. When the amount of PGDA is about 10%, the synergistic effects of diffusion and solvation are similar, and the reaction rate is the highest. In this way, it not only improves the application efficiency, but also ensures the total removal of $CO_2$ gas. In this study, the waterborne polyurethane coating was prepared and its good applicability and appearance were verified for carbon fiber metro vehicles.

**Author Contributions:** Conceptualization, Z.J.; methodology, C.L.; software, R.W.; validation, Z.Y.; formal analysis, Z.L.; investigation, R.W. and M.Z.; resources, M.Z.; data curation, R.W.; writing—original draft preparation, R.W.; writing—review and editing, Z.J. and C.L.; visualization, R.W.; supervision, Z.W.; project administration, Z.J.; funding acquisition, Z.J. All authors have read and agreed to the published version of the manuscript.

**Funding:** This work was supported by the National Key R&D Program of China (2016YFC0204400).

**Acknowledgments:** The authors also thank Frank Zhang and Steven Zhu from Covestro Co. for the discussions, which was helpful.

**Conflicts of Interest:** The authors declare no conflict of interest. The funders had no role in the design of the study; in the collection, analyses, or interpretation of data; in the writing of the manuscript, or in the decision to publish the results.

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
