# Peer review of "Study of the Effect of PGDA Solvent on Film Formation and Curing Process of Two-Component Waterborne Polyurethane Coatings by FTIR Tracking"

_coatings, doi:10.3390/coatings10050461_

Round 1
Reviewer 1 Report
The authors describe the impact of PGDA as an additive for polyurethane coatings, and claim a positive effect regarding microbubble inhibition.
In fact, I don't really understand the idea behind it: The most easiest way to inhibit formation of gas inclusions is to inhibit the gas formation by excluding water.
When small amounts of water couldn't be excluded from a practical point of view, reduction of the viscosity to improve the gas release appears to be an appropiate strategy. However, addition of 5 or 10% of this additive is from my point of view way too much:
The effect of the PGDA on the properties of the coating is not clearly described within this manuscript:
- When a monovalent alcohol is added to a polyisocyanate, it may affect the network formation of the polymer, which could imply drastic reduction of the mechanical stability (hardness, tensile strength, etc.).
- PGDA exhibits a boiling point of roughly 160 °C. What happens with the (unbound) PGDA?
Furthermore, the reaction kinetics of the curing process (IR spectra) show that in the basic formulation (0% PGDA) the isocyanate band at 2150 cm-1 is not significantly reduced, which makes me doubt the curing conditions (or the formulation). That the reduction of this band by addition of PGDA clearly indicates that the alcohol reacts with the isocyanate.
It is impressive that the authors painted a train with that formulation yielding good surfaces (by naked eye), but I am not convinced of the science behind it. Detailed investigation of the reaction and analysis of the properties of the coatings (e.g. tensile strength) could make the article acceptable, when these properties would not be affected too much by addition of the alcohol.
Author Response
Dear Reviewer & Editor:
Thank you very much for your kind letter, along with the constructive comments of the reviewer concerning our manuscript (coatings-769163). We have thoroughly considered all the comments of the reviewers and substantially revised our manuscript, and the major revised portions are marked in yellow in our revised manuscript. We also respond point by point to the reviewer’s comments as listed below, along with a clear indication of the location of the revision. We look forward to hearing from you.
Thanks again
Sincerely yours
Zhaohua Jiang
Response to Reviews
Question 1:
In fact, I don't really understand the idea behind it: The most easiest way to inhibit formation of gas inclusions is to inhibit the gas formation by excluding water.
Answer:
We are grateful to the reviewer for his suggestion. After we carefully thought about this question and read some relevant papers, we added some detailed description into that part to clearly show the reason of water added in the reaction in the revised manuscript (page 2: 1-7 line).
Question 2:
When small amounts of water couldn't be excluded from a practical point of view, reduction of the viscosity to improve the gas release appears to be an appropiate strategy. However, addition of 5 or 10% of this additive is from my point of view way too much:
The effect of the PGDA on the properties of the coating is not clearly described within this manuscript:
When a monovalent alcohol is added to a polyisocyanate, it may affect the network formation of the polymer, which could imply drastic reduction of the mechanical stability (hardness, tensile strength, etc.).
PGDA exhibits a boiling point of roughly 160 °C. What happens with the (unbound) PGDA?
Answer:
Thanks for the reviewer's suggestion, I added references to explain the normal amount of solvents such as PGDA adding in the waterborne polyurethane coating (page 2: 20-22 line).
Inspired by the reviewer, I added a description for the effect of the PGDA on the properties of the coating in the revised manuscript (page 2: 18-20 line).
The solvent containing hydroxyl group does have adverse effect on the coating performance, and the explanation has been added according to the reviewer's suggestion(page 2: 31-34 line).
In addition, the molecular structure formula of PGDA was added in the revision. Also, we added and rephrased the explanation to improve these parts (page 4: 1-3 line).
Question 3:
It is impressive that the authors painted a train with that formulation yielding good surfaces (by naked eye), but I am not convinced of the science behind it. Detailed investigation of the reaction and analysis of the properties of the coatings (e.g. tensile strength) could make the article acceptable, when these properties would not be affected too much by addition of the alcohol.
Answer:
This comment from reviewer is very relevant. In order to make the article more scientific, We deleted the figure of the train and added the statement that the coating performance can meet the requirements of CRRC(page 10: 27-28 line).

Reviewer 2 Report
This paper presents an interesting study of the effect of PGDA solvent on film formation and curing process of two component waterborne PU coatings.
The introduction allows the reader to well understand the purpose of this work. Nevertheless, it is would be desirable, in an international journal, to choose references that can be available for the reader. Here, the two references used (5 and 6) to introduce the role of the alcohol ether solvents are in Chinese and one of it is not available.
The major result of this paper comes from the study of the decrease of isocyanate IR band intensity along time, which is connected to the NCO reaction rate. This study clearly shows how this decrease depends on the % of PGDA.
IR study
- because of the volatilization of solvent and water during the experiment, it would be useful to have a “reference IR band” from an inert group used to normalize the data. Indeed, we can suppose that the %of PGDA could modify the volatilization rate of solvent and water during the experiment and thus, act on the peak absorbance independently of the consumption of NCO.
- Moreover, as mentioned in previous studies published on this precise subject, (for ex “Effects of Solvent Polarity on the Reaction of Phenol with Tolylene-2,4-Diisocyanate” Journal of Applied Polymer Science 123(1):580-584, doi:10.1002/app.34479) it is necessary to verify the linearity of the relationship Abs/concentration (and then use low concentration) to conclude on the progress of the curing reaction.
- At last, there is no indication about the number of experiments which have been done and uncertainties are not quoted.
NCO/OH molar ratio studied: in the formulation studied this ratio is indicated to be 1. Could the authors justify this choice because this molar ratio is often higher than 1 to avoid undesired NCO/H2O reaction as mentioned for example in ref “Waterborne polyurethanes,” Progress in Organic Coatings, Vol. 32, No. 1-4, 1997, pp. 131-136.) Moreover, it is not easy to determine the experimental ratio from the data indicated in tables 1 and 2. Could the author be more explicit please.
Presence of microbubbles: as indicated by the authors, the presence of microbubbles affect the quality of the coalesced film. However, this point is not documented in the paper (no image analysis, no quantification)
Role of PGDA as solvent-discussion: The authors don’t refer to numerous studies dealing with the effect of solvent for the urethan formation to enrich discussion. ( for example: doi.org/10.1002/pola.1987.080250919, Kinetics and mechanism of urethane reactions: Phenyl isocyanate–alcohol systems Journal of Polymer Science Part A: Polymer Chemistry, Effects of Solvent Polarity on the Reaction of Phenol with Tolylene-2,4-Diisocyanate, Journal of Applied Polymer Science 123(1):580-584, doi: 10.1002/app.34479…)
Author Response
Dear Reviewer & Editor:
Thank you very much for your kind letter, along with the constructive comments of the reviewer concerning our manuscript (coatings-769163). We have thoroughly considered all the comments of the reviewers and substantially revised our manuscript, and the major revised portions are marked in yellow in our revised manuscript. We also respond point by point to the reviewer’s comments as listed below, along with a clear indication of the location of the revision. We look forward to hearing from you.
Thanks again
Sincerely yours
Zhaohua Jiang
Response to Reviews
Question 1:
The introduction allows the reader to well understand the purpose of this work. Nevertheless, it is would be desirable, in an international journal, to choose references that can be available for the reader. Here, the two references used (5 and 6) to introduce the role of the alcohol ether solvents are in Chinese and one of it is not available.
Answer:
Thank you for your kind reminder. I have replaced the sixth reference as you suggested (page 2: 18-26 line).
Question 2:
IR study: because of the volatilization of solvent and water during the experiment, it would be useful to have a “reference IR band” from an inert group used to normalize the data. Indeed, we can suppose that the %of PGDA could modify the volatilization rate of solvent and water during the experiment and thus, act on the peak absorbance independently of the consumption of NCO.
Moreover, as mentioned in previous studies published on this precise subject, (for ex “Effects of Solvent Polarity on the Reaction of Phenol with Tolylene-2,4-Diisocyanate” Journal of Applied Polymer Science 123(1):580-584, doi:10.1002/app.34479) it is necessary to verify the linearity of the relationship Abs/concentration (and then use low concentration) to conclude on the progress of the curing reaction.
At last, there is no indication about the number of experiments which have been done and uncertainties are not quoted.
Answer:
Thank you for your reminder of spectrum consistency. According to the reviewer's suggestion, I added the references about the analysis of polyurethane resin infrared spectrum.(page 5: 17 line)
This paper recommended by the reviewer is very suitable for explaining the influence of polarity on chemical reaction, and has been quoted (page 8: 29 line ).
We have added a description of the detailed measures to eliminate uncertainty during the test process in this article (page 4: 13-15 line ).
Question 3:
NCO/OH molar ratio studied: in the formulation studied this ratio is indicated to be 1. Could the authors justify this choice because this molar ratio is often higher than 1 to avoid undesired NCO/H2O reaction as mentioned for example in ref “Waterborne polyurethanes,” Progress in Organic Coatings, Vol. 32, No. 1-4, 1997, pp. 131-136.) Moreover, it is not easy to determine the experimental ratio from the data indicated in tables 1 and 2. Could the author be more explicit please.
Answer:
We are grateful to the Reviewer for his professional suggestion. We have added a description of the NCO/OH molar ratio (page 4: 5-9).
In table 1 and table 2, we added the data of reaction group equivalent, which is convenient for readers to calculate.
Question 4:
Presence of microbubbles: as indicated by the authors, the presence of microbubbles affect the quality of the coalesced film. However, this point is not documented in the paper (no image analysis, no quantification)
Answer:
According to the reviewer's suggestion, we added the evaluation of the microbubble condition on the coating surface, and added the picture of microbubble under the optical microscope (page 7: 10-18 line).
According to the reviewer's suggestion, the expression of the effect of microbubbles on the film performance was added (page 2: 6-7 line).
Question 5:
Role of PGDA as solvent-discussion: The authors don’t refer to numerous studies dealing with the effect of solvent for the urethan formation to enrich discussion. ( for example: doi.org/10.1002/pola.1987.080250919, Kinetics and mechanism of urethane reactions: Phenyl isocyanate–alcohol systems Journal of Polymer Science Part A: Polymer Chemistry, Effects of Solvent Polarity on the Reaction of Phenol with Tolylene-2,4-Diisocyanate, Journal of Applied Polymer Science 123(1):580-584, doi: 10.1002/app.34479…)
Answer:
Thanks for the reference recommended by reviewers in this section, we think it is very suitable and have added. (page 8: 29 line).
Reviewer 3 Report
The manuscript by Wang and coworkers is about preparation of waterborne polyurethane coatings using polyhydroxyacrylate dispersion, polyisocyanate, and propylene glycol diacetate (PGDA). Authors used FTIR technique to study the formation of new bonds between OH and NCO groups occurring during film formation, and also studied curing processes by FTIR. The work has some merit for publication, but after major revision.
-in the abstract authors should put stress on the main outcome of this work. No data is presented to compare the performance of PGDA in the system in terms of values of responding variables. Also, there are several papers on this filed, hence the novelty of this work should be better explained.
-three works related to this work should be cited in the introduction:
https://www.sciencedirect.com/science/article/pii/S0300944018312669
https://www.sciencedirect.com/science/article/pii/S0021951719306116
https://onlinelibrary.wiley.com/doi/abs/10.1002/vnl.21414
-using FTIR technique for cure assessment is not exact. Why authors do not use calorimetric methods, based on DSC?
-I have a serious problem with 100% cure in figs. 6 & 7 after 15 min for all samples. It is strange and impossible. Even theoretically saying, gelation takes place at around 63%, where the system undergoes diffusion controlled curing.
-how the authors selected PGDA amount? We know that stoichiometry determines reaction and excess amount of reactants may cause complex formation in the system.
-figures 1-5 should be integrated into one fig. with a to e parts.
-figures 8, 9, and 10 should be integrated into a figure having a, b, and c parts. Also, fig. 10 should be deleted, which is not scientific.
-I cannot see any data in the conclusion for comparison. For example, the influence of PGDA amount on film formation and curing should be quantitatively addressed.
Author Response
Dear Reviewer & Editor:
Thank you very much for your kind letter, along with the constructive comments of the reviewer concerning our manuscript (coatings-769163). We have thoroughly considered all the comments of the reviewers and substantially revised our manuscript, and the major revised portions are marked in yellow in our revised manuscript. We also respond point by point to the reviewer’s comments as listed below, along with a clear indication of the location of the revision. We look forward to hearing from you.
Thanks again
Sincerely yours
Zhaohua Jiang
Response to Reviews
Question 1:
in the abstract authors should put stress on the main outcome of this work. No data is presented to compare the performance of PGDA in the system in terms of values of responding variables. Also, there are several papers on this filed, hence the novelty of this work should be better explained.
Answer:
According to the reviewer's suggestion, we added data to the summary to compare the performance of PGDA in the system (page 1: 19-20 line).
We are grateful to the Reviewer for his suggestion. We have added an explanation of the novelty of this study (page 3: 1-4 line).
Question 2:
three works related to this work should be cited in the introduction:
https://www.sciencedirect.com/science/article/pii/S0300944018312669
https://www.sciencedirect.com/science/article/pii/S0021951719306116
https://onlinelibrary.wiley.com/doi/abs/10.1002/vnl.21414
Answer:
We are grateful to the Reviewer for his suggestion. These three papers are very useful to enrich the discussion of the article, which we have quoted (References2,11 and 12).
Question 3:
using FTIR technique for cure assessment is not exact. Why authors do not use calorimetric methods, based on DSC?
Answer:
The instrument information of FTIR has been added according to the reviewer's suggestion(page 4: 21 line).
At present, there is no DSC instrument in our laboratory. Thank you for your recommendation. We will pay attention to the application of DSC in polyurethane coating research.
Question 4:
I have a serious problem with 100% cure in figs. 6 & 7 after 15 min for all samples. It is strange and impossible. Even theoretically saying, gelation takes place at around 63%, where the system undergoes diffusion controlled curing.
Answer:
Thank the censor from the bottom of my heart. The representation of PNCO has been confused in the calculation indeed. We have corrected Table 5, Figure 3 and Figure 4. Thank the reviewers again!
Question 5:
how the authors selected PGDA amount? We know that stoichiometry determines reaction and excess amount of reactants may cause complex formation in the system.
Answer:
We are grateful to the Reviewer for his suggestion. We have added a description about the selection of PGDA and the influence of the added amount (page 2: 31-38 line).
Question 6:
figures 1-5 should be integrated into one fig. with a to e parts.
Answer:
Thanks for the reviewer's suggestion. the Figures 1-5 have been integrated into one fig.
Question 7:
figures 8, 9, and 10 should be integrated into a figure having a, b, and c parts. Also, fig. 10 should be deleted, which is not scientific.
Answer:
Thanks for the reviewer's suggestion. We have combined figure 8 with figure 9, which is really clear and easy to understand; About figure 10, we added a description about it in the article.((page 8: 6 line)
Question 8:
I cannot see any data in the conclusion for comparison. For example, the influence of PGDA amount on film formation and curing should be quantitatively addressed.
Answer:
This comment from reviewer is very relevant. we added the comparison and increased the data of the effect of PDGA on the reaction degree when the dosage was 10% in the abstract and conclusion (page 1: 19-20 line and page 10: 37-38 line).
Round 2
Reviewer 1 Report
The manuscript has been significantly improved. In my opinion, the revised version can be accepted as it is.
Author Response
Dear Reviewer & Editor:
Thank you very much for your kind letter, along with the constructive comments of the reviewer concerning our manuscript (coatings-769163). We have thoroughly considered all the comments of the reviewers and substantially revised our manuscript, and the major revised portions are marked in yellow in our revised manuscript. We also respond point by point to the reviewer’s comments as listed below, along with a clear indication of the location of the revision. We look forward to hearing from you.
Thanks again
Sincerely yours
Zhaohua Jiang
Response to Reviews
For Reviewer #1:
Question 1:
The manuscript has been significantly improved. In my opinion, the revised version can be accepted as it is.
Answer:
Thank you from the bottom of my heart for your suggestions and comments to make our paper acceptable.
Reviewer 2 Report
Dear authors,
Thank you for your corrections.
However I have still some comments and some points have to be addressed:
Remark1: PGDA formula represented in figure 1, page 4 is not correct. Please correct it.
Remark 2: could you specify at which time correspond the pictures of the microbubbles under the optical microscope (page 7: 10-18 line)
Remark 3: In your answer to my question 5 you mentioned that you added references -(page 8: 29 line).- but I am not able to read them it in the new version.
However my major concern deals with IRFT analysis. As previously written, (question 2 in the first review) ) it is not so easy to quantitatively correlate the absorbance values you have measured with the reaction rate, as you did it in the paper.
For that purpose you need to refer to an internal standard. Indeed IR absorbance is not an absolute method to determine the concentration: NCO absorbance value also depends on the quality of the contact between the sample and the ATR crystal, on the water and solvent volatilization....
For example how do you explain the decrease of the absorbance of the peak at 2950-2850cm-1 (C-H alcane stretching) or of the peak at 1700-1740 cm-1 (carbonyl carbamate), this decrease is visible on the IRFT graphs containing 15%PGDA after 3 and 4h?
Morover it is necessary to verify the linearity of the relationship between NCO absorbance and NCO concentration in the range of concentrations studied in order to be able to quantitavely conclude on the progress of the curing reaction (doi:10.1002/app.34479) .
This issue has to be clarified.
Author Response
Dear Reviewer & Editor:
Thank you very much for your kind letter, along with the constructive comments of the reviewer concerning our manuscript (coatings-769163). We have thoroughly considered all the comments of the reviewers and substantially revised our manuscript, and the major revised portions are marked in yellow in our revised manuscript. We also respond point by point to the reviewer’s comments as listed below, along with a clear indication of the location of the revision. We look forward to hearing from you.
Thanks again
Sincerely yours
Zhaohua Jiang
Response to Reviews
Question 1:
PGDA formula represented in figure 1, page 4 is not correct. Please correct it.
Answer:
Thank you for your careful and professional examination. I have corrected the molecular formula of PGDA as required(page 4: 2 line).
Question 2:
Could you specify at which time correspond the pictures of the microbubbles under the optical microscope (page 7: 10-18 line)
Answer:
Thanks for the reviewer's suggestion, we have made clear the time of microbubble observation as required (page 8: 1 line).
Question 3:
In your answer to my question 5 you mentioned that you added references -(page 8: 29 line).- but I am not able to read them it in the new version.
Answer:
Thank the reviewer for the recommended reference. In the previous version, the serial number of this reference is [18], in this revision, the serial number is [13], which is quoted in lines 10-13 on page 3.
Question 4:
However my major concern deals with IRFT analysis. As previously written, (question 2 in the first review) ) it is not so easy to quantitatively correlate the absorbance values you have measured with the reaction rate, as you did it in the paper.
For that purpose you need to refer to an internal standard. Indeed IR absorbance is not an absolute method to determine the concentration: NCO absorbance value also depends on the quality of the contact between the sample and the ATR crystal, on the water and solvent volatilization....
For example how do you explain the decrease of the absorbance of the peak at 2950-2850cm-1 (C-H alcane stretching) or of the peak at 1700-1740 cm-1 (carbonyl carbamate), this decrease is visible on the IRFT graphs containing 15%PGDA after 3 and 4h?
Morover it is necessary to verify the linearity of the relationship between NCO absorbance and NCO concentration in the range of concentrations studied in order to be able to quantitavely conclude on the progress of the curing reaction (doi:10.1002/app.34479) .
This issue has to be clarified.
Answer:
Thanks for the reviewer's professional advice on "IRFT analysis". I read the references mentioned by the reviewer carefully and revised the analysis part of infrared spectrum in the article.(page 3: 10-13 line, page 4: 24-29 line, page 6: 25-29 line)
In addition, we carefully checked the English in the text and the mistakes including some grammar and spelling errors again. And all figures and the sequence of the references are renewed in the text.
Reviewer 3 Report
Tha quality of figure 2 is poor.
Author Response
Dear Reviewer & Editor:
Thank you very much for your kind letter, along with the constructive comments of the reviewer concerning our manuscript (coatings-769163). We have thoroughly considered all the comments of the reviewers and substantially revised our manuscript, and the major revised portions are marked in yellow in our revised manuscript. We also respond point by point to the reviewer’s comments as listed below, along with a clear indication of the location of the revision. We look forward to hearing from you.
Thanks again
Sincerely yours
Zhaohua Jiang
Response to Reviews
Question 1:
The quality of figure 2 is poor.
Answer:
Thank you for your suggestion. We have adjusted the quality of figure 2 as required (page 5-6).
Round 3
Reviewer 2 Report
Dear Authors,
Thank you for all your corrections.
I completely agree with using the ratio S'NCO /S'CH to normalize the NCO absorbance, this eliminates errors which can be due to a change of the quality of interface between the sample and the crystal used for ATR FTIR during the experiments for example.
However these are my two comments:
If PGDA evaporates during the experiments (4 hours) it will affect the intensity of SCH because alcane groups of PGDA also contributes to this signal. Could it be neglected?
I understand that you are maybe not able to do new experiments to verify the linearity of the relationship between NCO absorbance and NCO concentration in the range of the concentrations studied in your work. However to be able to quantitatively conclude on the progress of the curing reaction, you need to do preliminary tests as it has been published in different studies to validate-or not- this linear relationship (for example by using different initial NCO ratios for the same formulation and then verify that the initial SNCO /SCH linearly varies with the initial NCO concentration tested).
Formula 5: the minus sign was left.
Author Response
Dear Reviewer & Editor: Thank you very much for your kind letter, along with the constructive comments of the reviewer concerning our manuscript (coatings-769163). We have thoroughly considered all the comments of the reviewers and substantially revised our manuscript, and the major revised portions are marked in blue in our revised manuscript. We also respond point by point to the reviewer’s comments as listed below, along with a clear indication of the location of the revision. We look forward to hearing from you. Thanks again Sincerely yours Zhaohua Jiang Response to Reviews Question 1: If PGDA evaporates during the experiments (4 hours) it will affect the intensity of SCH because alcane groups of PGDA also contributes to this signal. Could it be neglected? Answer: Thanks for your serious review and rigorous suggestion, I fully agree that it is necessary to add a discussion on the volatilization of PGDA in this paper (page 4: 28-32 line). Question 2: I understand that you are maybe not able to do new experiments to verify the linearity of the relationship between NCO absorbance and NCO concentration in the range of the concentrations studied in your work. However to be able to quantitatively conclude on the progress of the curing reaction, you need to do preliminary tests as it has been published in different studies to validate-or not- this linear relationship (for example by using different initial NCO ratios for the same formulation and then verify that the initial SNCO /SCH linearly varies with the initial NCO concentration tested). Answer: Thank you for your professional guidance in this field! If I do some experiments according to your suggestions, it will make this article more excellent indeed. But it's a pity that now and in the foreseeable weeks, I can't go back to the school lab for experiments due to the epidemic, so I can only cite some previous research results to discuss. I beg you to understand more. (page 4: 34-38 line and page 5: 1-2 line ). Question 3: Formula 5: the minus sign was left. Answer: Thank you for your careful and professional examination. I have corrected the mistake (page 7: 3 line).